# Nature of the Metal Insulator Transition in High-Mobility 2D_Si-MOSFETs

**DOI:** 10.3390/nano13142047

**Published:** 2023-07-11

**Authors:** F. Elmourabit, S. Dlimi, A. El Moutaouakil, F. Id Ouissaaden, A. Khoukh, L. Limouny, H. Elkhatat, A. El Kaaouachi

**Affiliations:** 1Laboratory of Sciences and Technologies of Information and Communication (LSTIC), Microelectronics, Microwaves, Instrumentation and Information (MM2I), Department of Physics, Faculty of Sciences, Chouaib Doukkali University, Av. des Facultés, El Jadida 24000, Morocco; 2Department of Electrical and Communication Engineering, United Arab Emirates University, Al Ain P.O. Box 15551, United Arab Emirates; 3Equipe des Energies Nouvelles et Ingénierie des Matériaux (ENIM), Laboratoire de Sciences et Techniques de L’ingénieur (LSTI), Physics Department, Faculty of Sciences and Technics Errachidia, Moulay Ismail University, Meknes 50050, Morocco; 4Electrical Engineering Department, National School of Applied Sciences of Tangier (ENSAT), University of Abdelmalek Essaadi, B.P. 416, Tangier 93000, Morocco; 5Department of Physics, Faculty of Sciences, Ibn Zohr University, B.P. 1136, Agadir 80000, Morocco

**Keywords:** percolation-type MIT, electrical conductivity, carrier density, 2D Si-MOSFETs

## Abstract

Our investigation focuses on the analysis of the conductive properties of high-mobility 2D-Si-MOSFETs as they approach the critical carrier density, $n_{sc}$ (approximately $0.72\times 10^{11}\;{cm}^{-2}$), which marks the metal insulator transition (MIT). In close proximity to the $n_{sc}$, the conductivity exhibits a linear dependence on the temperature (*T*). By examining the extrapolated conductivity at the absolute zero temperature (*T* = 0), denoted as $\sigma_{0}$, as a function of the electron density $n_{s}$, we identify two distinct regimes with varying $\sigma_{0}(n_{s})$ patterns, indicating the existence of two different phases. The transition from one of these two regimes to another, coinciding with $n_{sc}$, is abrupt and serves as the focus of our investigation. Our aim is to establish the possibility of a percolation type transition in the 2D-Si-MOSFETs’ sample. In fact, we observed that the model of percolation is applicable only for densities very close to $n_{sc}^{*}=n_{2}$ (where $n_{2}$ is the linear extrapolation of $\sigma_{0}$), indicating the percolation type transition essentially represents a phase transition at the zero temperature.

## 1. Introduction

A two-dimensional (2D) percolation-type metal insulator transition (MIT) has been a topic of significant research interest in recent years. Several studies have explored the behavior of conductivity and the transition from a conducting state to an insulating state in 2D systems [1,2,3]. One notable study by Kravchenko et al. from 1994 to 2001 [4,5,6] observed the MIT in Si samples, marking an important milestone in the field. Since then, researchers have conducted extensive investigations on various types of samples to further understand the nature of the 2D MIT. For instance, studies have been carried out on Si-MOS (Si-Metal Oxide Semiconductor) samples [7], p-Si/SiGe [8], n-Si/SiGe [9], p-AlGaAs [10], n-AlGaAs [11], and n-AlAs [12] samples, each shedding light on the characteristics of the MIT in different materials. These investigations have revealed intriguing observations that challenge the predictions of the scale theory proposed by Abrahams et al. [13], which suggested the absence of the MIT in 2D and 1D systems. In order to elucidate the underlying mechanisms of the 2D-MIT, several theories have been put forth. Certain theories emphasize the role of disorder and electron–electron (e-e) interactions [14,15,16,17,18], whereas others propose spin-orbit diffusion as a potential contributing factor [6,19]. Recent research has particularly delved into the effects of e-e interactions on the metallic side of the MIT [20,21,22,23]. Several experimental studies in the literature suggest that the 2D-MIT is a percolation-type transition. However, the parameter space investigated in this context, including the temperature, density, and mobility, has been relatively limited compared to the numerous claims advocating the prevailing viewpoint that the 2D-MIT is a quantum phase-type transition. Most of these studies focused on two-dimensional systems of holes and rarely on two-dimensional systems of electrons. The system studied in this paper is of a good quality (cleaner) because of its high mobility and therefore it is less disordered. This paper presents compelling evidence that supports the proposition that the MIT observed in 2D Si MOSFETs can be characterized as a percolation-type transition. Indeed, we conducted a re-analysis of the data obtained by Pudalov and colleagues [24]. We examined the Si-15 sample, which encompasses different carrier densities $n_{s}$ within the range $0.449-4.98\times 10^{11}\;cm^{-2}$. The peak mobility of this sample measures at $\mu =4.1\;m^{2}/Vs$. The conductivity measurements were conducted using a ^3^He-cryostat, reaching temperatures as low as 0.3 K. The four-terminal AC technique was employed, wherein lock-in amplifiers detected signals within the frequency range of 13 to 17 Hz.

## 2. Results and Discussion

In Figure 1, we plotted the electrical conductivity σ (σ=1/ρ) against the temperature T for the sample Si-15, for several electron densities $n_{s}$ near the MIT on both metallic and insulating sides in the range of $0.503\;\times 10^{11}\;cm^{-2}<n_{s}<1.2\;\times 10^{11}\;cm^{-2}$. Insulating behavior is observed for densities below $n_{sc}=0.719\times 10^{11}\;cm^{-2}$, where the critical value of electron density, $n_{sc}$, serves as the threshold that distinguishes the metallic and insulating regions. On the insulating side, the conductivity decreases rapidly as the temperature is reduced. On the higher carrier density side, metallic behavior is observed, where the conductivity diminishes with the decreasing temperature [24].

In the present work, our investigation focuses on the behavior of the electrical conductivity σ on both sides of the MIT near the critical electron density $n_{sc}$. This critical density serves as the demarcation point that separates the metallic and insulating regions of the MIT. In analyzing Figure 1, we notice that around the critical density $n_{sc}$, the curves representing σ versus T are almost linear for T>1.7 K.

To illustrate this linear $\sigma \left(T\right)$ temperature dependence in the vicinity of $n_{sc}$, we demonstrate linear fits for several densities using the expression:(1)$$\sigma \left(T\right)=\sigma_{0}+\beta T$$
where $\sigma_{0}$ is the linear extrapolation to the zero temperature (*T* = 0) and $\beta =\frac{d\sigma }{dT}$ is the conductivity slope.

The conductivity appears to be independent of temperatures below 2.1 K. The percolation model adopted is only valid for densities very close to the critical density (outliers at very low temperatures are few in number compared with the total number of experimental points). This aberration is incomprehensible and unexpected. We suspect the appearance of quantum phenomena at very low temperatures. That is why we only consider linear points.

In Figure 2, we plotted $\sigma_{0}$ versus $n_{s}$ and it becomes apparent that there are two distinct regimes characterized by different linear relationships between $\sigma_{0}$ and $n_{s}$. In the low-density insulating phase, we observe that $\sigma_{0}$ increases as the density increases, with a rate of $0.6\;e^{2}/h$ per $10^{11}\;cm^{-2}$. This implies that for every increase of $10^{11}$ charges per square centimeter, the conductivity increases by $0.6\;e^{2}/h$. The slope is multiplied by four and becomes $2.5\;e^{2}/h$ per $10^{11}\;cm^{-2}$ on the high-density side. Remarkably, the conductivity $\sigma_{0}\left(n_{s}\right)$ exhibits a distinct inflection point where the two linear trends intersect, occurring at approximately $n_{s}\approx 0.72\times 10^{11}\;cm^{-2}$, which coincides with the critical density $n_{sc}$ of the 2D-MIT. And the σ intercept $\sigma_{0}$ at this density equals exactly $0.105\;e^{2}/h$. This value corresponds to the quantum conductance $e^{2}/h$ divided by 10 per square. In both phases, we extend the linear $\sigma_{0}\left(n_{s}\right)$ relationships to estimate the point where conductivity reaches zero. On the low-density side, this extrapolation leads us to a value of $n_{1}=0.52\times 10^{11\;}cm^{-2}$. Similarly, on the high-density side, the extrapolation brings us to a value of $n_{2}=0.68\times 10^{11}\;cm^{-2}$ where conductivity is projected to reach zero.

In Figure 3, the slope β in Equation (1) is represented as a function of the carrier density $n_{s}$. It remains relatively unchanged for $n_{s}<0.72\times 10^{11}\;cm^{-2}$ (0.18<β<0.2 $\pm \;0.01\;e^{2}h^{-1}K^{-1}$), and rapidly decreases for higher densities ($n_{s}>0.72\times 10^{11}\;cm^{-2}$). Previous studies [25,26] have reported the observation of a linear σ(T) relationship in p-type GaAs samples. In these studies, the observed β values ranged from approximately 3 to 5 $e^{2}/h$ per Kelvin in p-type GaAs samples. This is in contrast to the significantly smaller values of approximately 0.18 to 0.2 $e^{2}h^{-1}K^{-1}$ observed in the studied Si quantum well sample. The notable difference of approximately one order of magnitude between these values may be attributed to the distinct Fermi temperatures ($T_{F}$) in these two electronic systems.

In previous works [25,26], the authors showed that for $T\leq T_{F}$, β  is ~$1/T_{F}$ (β=dσ/dT ), where $T_{F}$ is the Fermi temperature given by:(2)$$T_{F}=E_{F}/k_{B}$$
where $k_{B}$ is the Boltzmann’s constant, and $E_{F}$ is the Fermi energy given by:(3)$$E_{F}=\pi \hbar^{2}n_{S}/2m^{*}$$

The value of $m^{*}$ is assumed to be the same for all temperatures. $m^{*}$ is calculated with the formula $m^{*}\approx 1+0.058r_{s}+0.052r_{s}^{2}$, which provides a reliable estimate for the effective mass obtained from Shubnikov–de Haas oscillations [27], where $r_{s}=1/a_{B}{\left(\pi n\right)}^{1/2}$ is the interaction parameter and $a_{B}$ is the effective Bohr radius [28,29].

Figure 4 depicts the plot of $T_{F}$ against $n_{S},$ utilizing Equations (2) and (3). Notably, we observed that the range of $3\;K<T_{F}<6.5\;K$ in our study is approximately 25 times larger than the corresponding range (around $10^{-1}$ to ${4\times 10}^{-1}$) observed in 2D p-GaAs samples. Consequently, the rate of change of conductivity with respect to the temperature (dσ/dT) in Si-15 is anticipated to be 25 times smaller than that in p-GaAs. It is worth noting that, in our case, the temperature T always remains lower than $T_{F}$.

Theoretical investigations have also explored the linear relationship between σ(T) and the temperature in a high-temperature regime. In reference [27], the authors conducted calculations using a percolating system considering screened charged impurity scattering. They observed that for temperatures significantly larger than the Fermi temperature $T_{F}$ ($T>>T_{F}$), the conductivity, σ(T), followed a proportionality of $\sigma (T)\;\propto \;T/T_{F}$, indicating a linear dependence on the temperature.

In addition to this model of screening, the microemulsion model proposed in Refs. [28,30] offers an explanation for the linear σ(T) behavior at high temperatures. However, neither of these models can quantitatively account for the experimental observation that dσ/dT is relatively not dependent on the carrier density around the $n_{sc}$. According to the model of screening that depends on the temperature [30], dσ/dT is expected to be proportional to $1/T_{F}$ or $n_{s}^{-1}$. Consequently, as the 2DES density increases from 0.503 to $1.2\times 10^{11}\;cm^{-2},\;$ the slope is expected to decrease by a factor of approximately 1.66. On the other hand, the microemulsion model suggests a relationship of $d\sigma /dT\sim n_{s}^{2}$ [31], which would lead to a slope change of roughly 5.6 in the same density range. Zala et al. [32] demonstrated a strong density dependence of the slope dσ/dT in a low-density regime, whereas we obtained a nearly constant value (0.18<β<0.2$\;\pm 0.01\;e^{2}/h\;per\;Kelvin$). Currently, the cause of this inconsistency between experimental results and theoretical predictions remains unknown.

Next, let us focus on the MIT observed in the Si-15 sample. The conductivity of the 2DES in this case can be described using the percolation model. According to this model, the conductivity follows a scaling function represented as $\sigma \sim {\left(n_{S}/n_{p}-1\right)}^{\alpha }$, where $n_{S}$ and $n_{p}\;$ are the electron and hole density, respectively. Within the framework of a classical percolation-type transition, the exponent α is determined to be 4/3 based on references [2,3,28,33]. Figure 5 illustrates the measured conductivity (σ) at a temperature of T=0.3 K plotted against $\left(n_{s}/n_{p}-1\right)$ on a logarithmic scale. Due to the unknown value of $n_{p}$ derived from finite temperature measurements, three different trial densities ($n_{1}$, $n_{2}$, and $n_{sc}$) are utilized, expressed in units of $10^{11}\;cm^{-2}$. The value $n_{1}=0.52$ is obtained by extrapolating to $\sigma_{0}=0$ from the low-density regime, while $n_{2}=0.68$ corresponds to extrapolation from the high-density regime. Additionally, $n_{sc}=0.719\;\times \;10^{11}\;cm^{-1}$ represents the critical density of the remarked 2D-MIT. For $n_{2}=0.68\times 10^{11}\;cm^{-2}$, a satisfactory linear fit is achieved with an α value of 1.29. This value is close to the classical model’s value of 4/3 and is obtainable over a range of two decades in σ. We observed that the deviation from the percolation fitting described above coincides with the density at which a noticeable “kink” is observed in Figure 3. Based on this observation, it can be inferred that the electronic phase in the low-density regime exhibits differences compared to the high-density regime. Regarding the density $n_{p}=n_{2}$, it remains uncertain at this point whether this correlation is coincidental or if the two densities are deeply interconnected. Conversely, when fitting to a power-law behavior with $n_{p}=n_{1}$ or $n_{sc}$, the results are not satisfactory. Our observations when applying the percolation model to the Si-15 sample indicate that the power-law conductivity behavior in this model holds only for densities that are in close proximity to $n_{sc}^{*}$, and the percolation transition essentially represents a phase transition at T=0.

In previous studies [3,34,35,36], we made an observation that the metallic state remains intact within a range of conductivity up to $n_{s}=0.719\;\times \;10^{11}\;cm^{-2}$ and at temperatures below 3 K in the Si-15 sample mentioned earlier. It is worth noting that the logarithmic dependence of conductivity on the temperature is relatively weak in this case, resulting in an increase in the electrical conductivity of the two-dimensional metal as the temperature (T) decreases. This logarithmic temperature dependence provides compelling evidence for the quantum nature of the two-dimensional metallic state as the temperature decreases. Additionally, we demonstrated the presence of a delocalizing logarithmic correction to the resistivity, which contributes to the overall prefactor ($C_{T}$) in the expression. The prefactor $C_{T}$ can be expressed as the sum of two prefactors dependent on disorder, each having opposite signs. Specifically, we can represent it as $C_{T}=C_{deloc}+C_{wloc}$. The positive one, $C_{wloc}$, represents the quantum interference of a single particle (“backscatter”); however, the negative one, $C_{deloc}$, represents the contribution of the interaction between carriers.

We also showed a temperature *T*^*^ according to which the expression of the electrical resistivity can be represented by either one of the expressions mentioned below: ($\rho =\rho_{0}+\rho_{1}\exp\left(-{\left(T_{0}/T\right)}^{p}\right)$ at $T>T^{*}$ and $\rho =\rho_{0}-\rho_{0}^{2}C_{T}\ln\left(T/T^{*}\right)$ at $T<T^{*}$). *T^*^* depends on carrier density $n_{s}$ and varies between 0 and 1 K for $n_{s}$ in the range $1.2-4.98\times 10^{11\;}cm^{-2}$ in the sample Si-15.

The percolation transition occurs due to the existence of a disorder potential in the 2D gas, which possesses a characteristic length greater than the average interparticle distance. In high-density regimes, where the surface density $n_{s}\;$ is sufficiently large, there are enough carriers to occupy the potential wells, leading to the smoothing out of the effective disorder and the system becoming homogeneous. However, at lower densities, the screening of disorder by carriers becomes less effective, resulting in the emergence of density inhomogeneities: carriers preferentially occupy the potential wells while the “peaks” are empty, as illustrated in Figure 6. If the depopulated regions are numerous enough, they prevent the aggregation of carrier-rich regions and thus the conduction of the current (at least at the finite temperature, through a temperature-dependent disorder screening mechanism). Therefore, there exists a percolation threshold characterized by a critical density, denoted as $n_{sc}^{*}$, above which the system exhibits metallic behavior, and below which it displays insulating properties.

Several authors have proposed phenomenological percolation models to describe the metal insulator transition. Meir’s non-interacting model [37] describes transport in a 2D gas using a classical network of quantum resistors, representing tunneling events connecting populated carrier regions across depopulated regions. Another notable model is Shi et al.’s meta-percolation model [38], which is a quantum percolation model in a square lattice with finite phase coherence. Transport through the links occurs coherently, with a phase decoherence introduced in a certain fraction of these links. In real systems, this type of percolation may arise from a mixture of two phases with different densities, where the high-density phase represents an electron liquid and the low-density phase corresponds to a Wigner crystal or glass.

The percolation model proposed by He and Xie describes a system consisting of a conductive liquid phase separated by regions of an insulating vapor phase [39]. In the case of Si-MOSFETs, the negative charges of the electron gas are compensated by mobile positive charges from the gate. Thus, the system can be considered as two interacting carrier gases: the 2D electron gas and a hole gas. At a high electron density, the system behaves like a metallic electron-hole liquid, while at a low density, it behaves as an insulating exciton gas. At an intermediate density, there is a mixture of these two phases.

Shi et al. [39] proposed that an electron gas with $r_{s}>2$ becomes unstable, and it is favorable for it to reduce its surface area in order to increase rs. This leads to the formation of dense liquid droplets surrounded by empty regions (or regions of a lower density, forming a disordered Wigner crystal when the temperature is non-zero). However, this reduction in surface area is accompanied by an increase in Fermi energy. They demonstrated that, in a system without disorder, the mechanism of droplet formation is unfavorable. On the other hand, in the presence of disorder, electrons tend to gather in potential minima, promoting droplet formation. They further showed that the Coulomb interaction needs to be screened for the formation of these droplets, which is the case in experimental systems, either due to a metallic gate in silicon MOSFETs or through the Thomas–Fermi mechanism in heterostructures.

It is worth noting that the model developed by Spivak [40] to interpret the behavior of resistivity in the metallic phase leads to a similar scenario of percolation, where a metallic Fermi liquid percolates within an insulating Wigner crystal. Spivak also considers the formation of liquid regions mixed with crystallites, which are stabilized by the screened Coulomb interaction between electrons. These various models are all based on the presence of density inhomogeneities in the metallic phase.

CMOS technology has been pivotal in revolutionizing the semiconductor industry and driving the digital revolution. However, with CMOS devices approaching their physical limits, researchers have focused on advancing CMOS technology to overcome challenges and expand its boundaries [41,42,43,44,45]. Nanowire transistors leverage ultra-thin nanowires to improve electrostatic control and short-channel behavior, enabling further scaling. Gate-All-Around (GAA) transistors exploit three-dimensional channel control, enhancing electrostatics and reducing leakage currents for improved energy efficiency. High-mobility channel materials (e.g., germanium, III-V compounds) harness superior electron and hole mobility to achieve higher device speeds and improved switching performance. 2D materials (e.g., graphene, transition metal dichalcogenides) explore atomically thin materials’ unique properties, enabling ultra-thin, flexible, and highly efficient CMOS devices. Spintronics harness electron spin for information storage and processing, potentially enabling low-power, non-volatile, and highly scalable computing. Quantum computing leverages the principles of quantum mechanics to revolutionize computing, with potential applications in cryptography, optimization, and material science. Heterogeneous integration seamlessly combines diverse materials, devices, and technologies on a single chip, facilitating efficient data flow, improved performance, and reduced power consumption. Advanced packaging techniques such as 3D integration, wafer-level packaging, and chiplet architectures overcome interconnect limitations, enabling higher performance systems.

Advancements in abridging CMOS technology have resulted in remarkable breakthroughs, addressing limitations and paving the way for future innovations. By exploring novel transistor architectures, leveraging advanced materials, exploring alternative computing paradigms, and embracing system-level integration, researchers are pushing CMOS technology boundaries. These advancements hold immense potential for achieving higher performance, improved energy efficiency, enhanced reliability, and unprecedented computational capabilities. Staying informed about the advancement enables researchers and industry professionals to leverage these advancements, driving further progress, and shaping the future of CMOS technology.

## 3. Conclusions

In summary, the Si-MOSFETs’ sample exhibits a significant electron mobility near the MIT in a 2D system. The conductivity shows a linear variation with the temperature for $T\leq T_{F}$ on both sides of the MIT and around the critical density $n_{sc}$. When plotting $\sigma_{0}$ against the carrier density $n_{s}$, a graphical representation reveals two linear regimes intersecting at the critical density $n_{sc}$, corresponding to the electronic phases in the low-density and high-density regimes, respectively. Furthermore, it is observed that the model of percolation is applicable only for densities very close to $n_{sc}$ (where $n_{sc}^{*}=n_{2}$), indicating that the percolation type transition essentially represents a phase transition at the zero temperature. The percolation scenario provides a realistic description of disorder, but it tends to downplay the significance of interaction effects, which actually play a crucial role in the system. While the percolation model may be applicable in describing the metal insulator transition, it does not contribute much to understanding the underlying origin of the metallic behavior [3].

## Figures and Tables

**Figure 1 nanomaterials-13-02047-f001:**
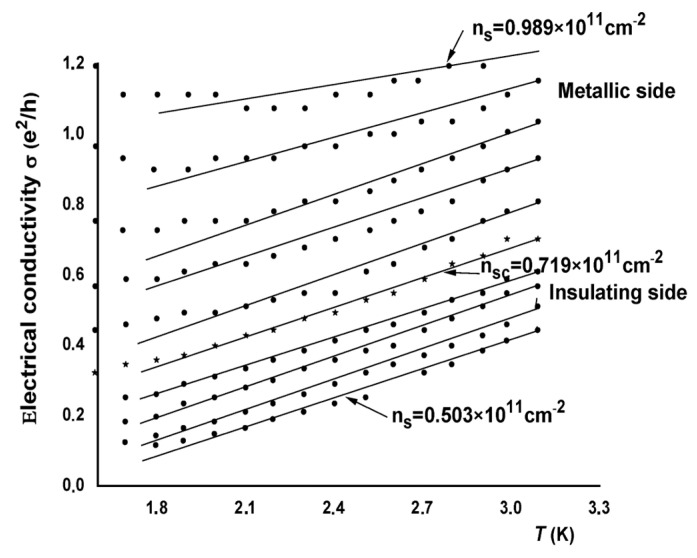
Electrical conductivity σ as a function of the temperature *T* in the vicinity of the critical density $n_{sc}$ for sample Si-15 for different electron densities from $0.503\;to\;0.989\;\times 10^{11}\;cm^{-2}$. Figure adapted from Ref. [24].

**Figure 2 nanomaterials-13-02047-f002:**
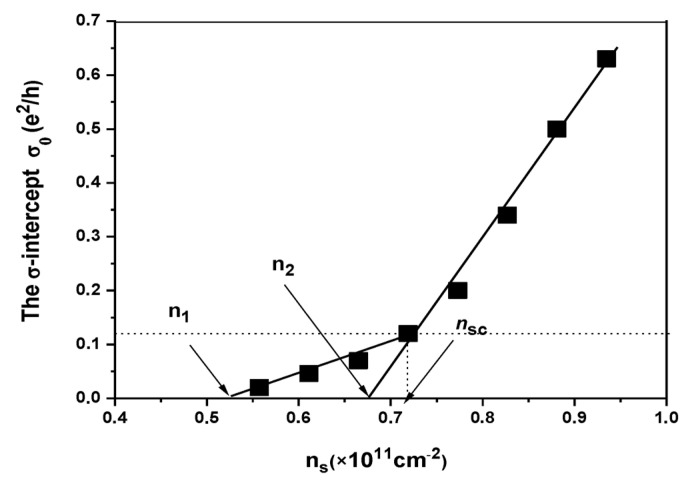
Conductivity $\;\sigma_{0}(n_{s})$ plot, and the extrapolated $\sigma_{0}\;$ values from the linear fits are plotted accordingly. The linear fits extend to the points where $\sigma_{0}\;$ is predicted to reach zero conductivity, which are $n_{1}=0.52\times 10^{11}\;cm^{-2}$ for the low-$n_{s}$ regime and $n_{2}=0.68\times 10^{11}\;cm^{-2}\;$ for the high-$n_{s}$ regime.

**Figure 3 nanomaterials-13-02047-f003:**
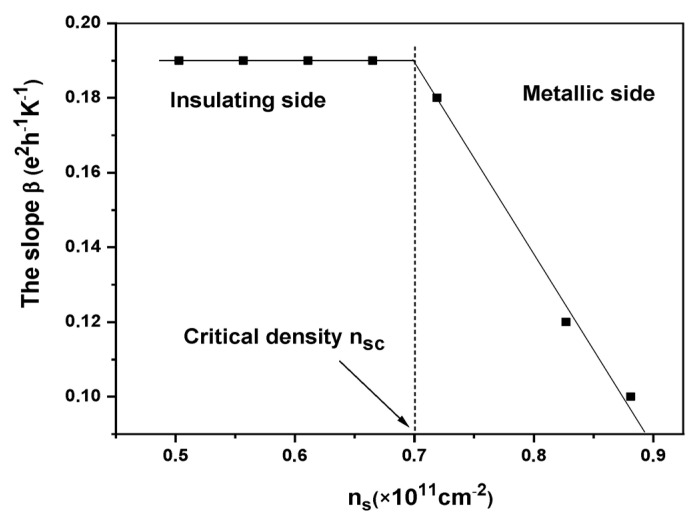
The slope β in Equation (1) as a function of carrier density $n_{s}$.

**Figure 4 nanomaterials-13-02047-f004:**
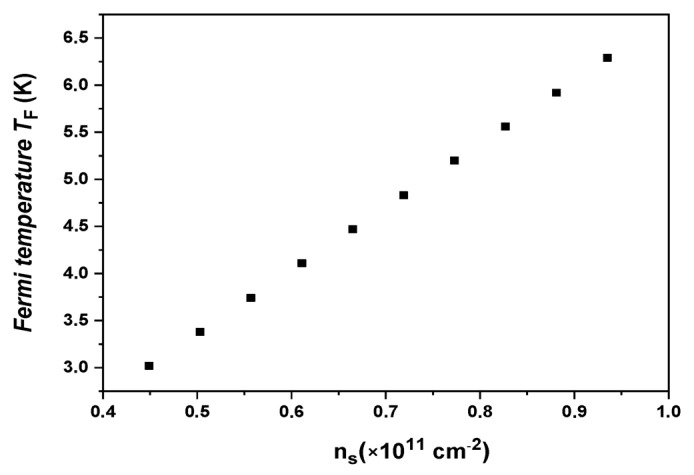
Fermi temperature $T_{F}\;$ as a function of carrier densities $n_{s}$.

**Figure 5 nanomaterials-13-02047-f005:**
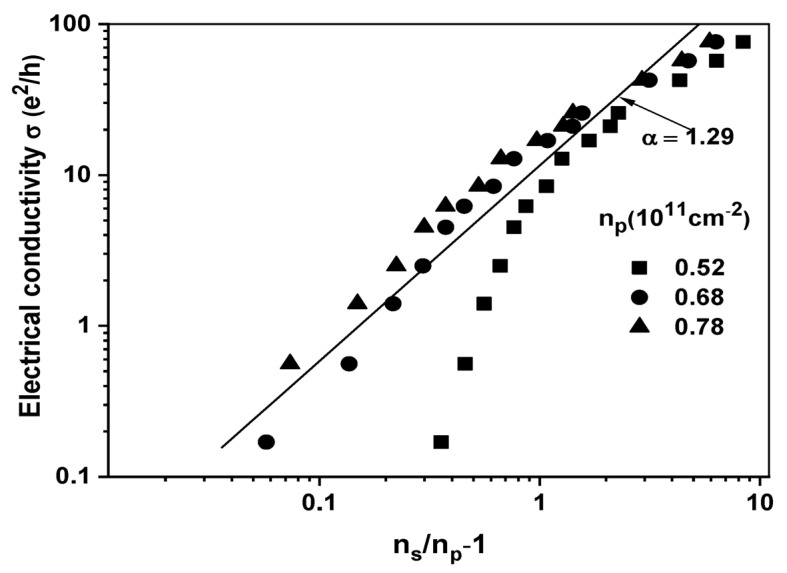
Percolation plot of $\sigma \left(T=0.3\;K\right)$ in a log–log scale.

**Figure 6 nanomaterials-13-02047-f006:**
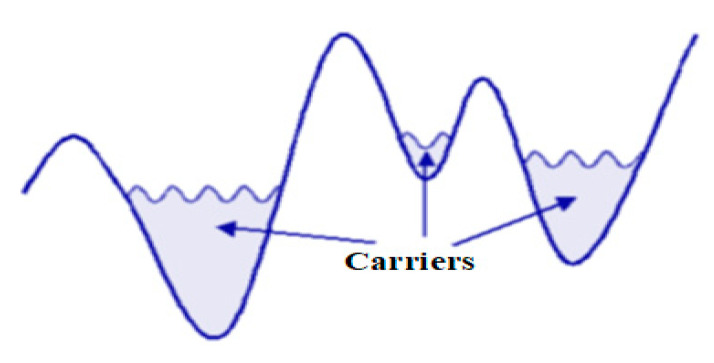
Density inhomogeneity in a long-range disorder potential.

## Data Availability

Data will be made available on request.

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
