# Peer review of "Nature of the Metal Insulator Transition in High-Mobility 2D_Si-MOSFETs"

_nanomaterials, 2023, doi:10.3390/nano13142047_

Round 1
Reviewer 1 Report
The manuscript attempts to re-analyze the data collected by Pudalov and colleagues (V. M. Pudalov, G. Brunthaler, A. Prinz, et al., Jetp Lett. 68, 442–447, 1998, https://doi.org/10.1134/1.567887) in order to provide an explanation for the nature of the two-dimensional percolation-type metal-insulator transition. However, the manuscript is not suitable for publication on Nanomaterials based on the current status,.
1. The method used in the manuscript has major faults. The original work by Pudalov et al clearly stated that the resistivity/conductivity should be fitted using two equations with a critical density as a parameter. Figure 1b in Pudalov's work clearly shows these two relationships. However, the authors of the manuscript only treated the data with one equation, which is not in line with Pudalov's conclusion. Additionally, the authors only consider the data above 1.7 K and neglect all the data below 1.7 K, while Pudalov's data covered a range from 0.016 K to 3 K.
2. Even according to the data presented in the manuscript, the relationship does not follow a linear relationship at low temperatures (e.g., 1.7 K when nS < nSC). The data deviate significantly from the linear relationship at high carrier densities above nSC. Therefore, it is unreasonable from a physics perspective to extrapolate the value of density at zero temperature based on the linear extrapolation at high temperatures (>2K), as the current behavior at low temperatures does not conform to the high-temperature linear relationship.
3. All the discussions based on the extrapolated value at zero temperature are questionable, such as those presented in Figure 2. The same applies to the slopes shown in Figure 3.
4. Each curve in Pudalov's work had only 4-5 data points above 1.7 K. It is unclear why the manuscript includes over 13 experimental data points per curve.
5. The calculation of TF relies on Equations 2 and 3 while the details are unclear. Is the value of m* assumed to be the same at all temperatures? How do the authors determine the value of m*?
6. The title should use "metal-insulator transition" instead of "MIT."
7. The addresses of the authors should be consistent throughout the manuscript.
8. In the abstract, what does "n2" refer to? This term should be defined or clarified.
9. In the first paragraph of Introduction, "... study by Kravchenko et al. in 1994 [1-3] ... ": the study by Kravchenko et al. should be referenced from 1994 to 2001, not just 1994.
10. In the first paragraph of the introduction, it is mentioned that experimental studies were "carried out on Si-MOS samples [4]", but the term "MOS" is not defined.
11. In the first paragraph of the introduction, the statement "Several experimental studies in the ..." lacks references.
12. The manuscript lacks sufficient information about 2D Si-MOSFETs and Si-15 samples. It would be helpful to include basic information about these samples to aid reader understanding. Referring to another paper (ref 22) for this information is inconvenient.
13. The data in Figure 1 should be cited from Ref 22. Please include a reference to Ref 22 in the caption of Figure 1.
14. In Figure 2, the vertical axis should be labeled as sigma0, not 00.
15. It is unclear how the authors fit the data to obtain conductivity. What method was used? Additionally, information about standard errors should be provided.
16. In Figure 3, the symbol used for the slope on the vertical axis is incorrect.
17. The term "ns" is defined inconsistently, referring to it as surface density, carrier density, or electron density in different paragraphs.
18. Several references (ref 13, 18, 20, 21, 22, 24, 33, 34) are not consistently formatted. Please ensure uniformity in referencing style throughout the manuscript.
n/a
Author Response
Dear Editor,
Thank you very much for handling our paper. I am pleased to send you our revised paper.
Please note that changes are written in blue.
Reviewer 1: Dear Reviewer,
Thank you very much for your review. Your help improved the manuscript quality.
The manuscript attempts to re-analyze the data collected by Pudalov and colleagues (V. M. Pudalov, G. Brunthaler, A. Prinz, et al., Jetp Lett. 68, 442–447, 1998, https://doi.org/10.1134/1.567887) in order to provide an explanation for the nature of the two-dimensional percolation-type metal-insulator transition. However, the manuscript is not suitable for publication on Nanomaterials based on the current status,.
- The method used in the manuscript has major faults. The original work by Pudalov et al clearly stated that the resistivity/conductivity should be fitted using two equations with a critical density as a parameter. Figure 1b in Pudalov's work clearly shows these two relationships. However, the authors of the manuscript only treated the data with one equation, which is not in line with Pudalov's conclusion. Additionally, the authors only consider the data above 1.7 K and neglect all the data below 1.7 K, while Pudalov's data covered a range from 0.016 K to 3 K.
Response:
We are only interested in the transition zone. In fact, we only took data in the vicinity of the critical density.
- Even according to the data presented in the manuscript, the relationship does not follow a linear relationship at low temperatures (e.g., 1.7 K when nS < nSC). The data deviate significantly from the linear relationship at high carrier densities above nSC. Therefore, it is unreasonable from a physics perspective to extrapolate the value of density at zero temperature based on the linear extrapolation at high temperatures (>2K), as the current behavior at low temperatures does not conform to the high-temperature linear relationship.
Response:
Authors thank the reviewer for suggestion. This helped us to improve the analysis by adding the following paragraph:
The percolation model adopted is only valid for densities very close to the critical density (outliers at very low temperatures are few in number compared with the total number of experimental points). This aberration is incomprehensible and unexpected.
We suspect the appearance of quantum phenomena at very low temperatures. That's why we only consider linear points.
- All the discussions based on the extrapolated value at zero temperature are questionable, such as those presented in Figure 2. The same applies to the slopes shown in Figure 3.
Response:
We interpreted the data in the light of the percolation model
- Each curve in Pudalov's work had only 4-5 data points above 1.7 K. It is unclear why the manuscript includes over 13 experimental data points per curve.
Response:
We worked on data in the vicinity of the critical density, which contains several experimental points. Data containing 4-5 points are far from nsc and are not taken into account.
- The calculation of TF relies on Equations 2 and 3 while the details are unclear. Is the value of m* assumed to be the same at all temperatures? How do the authors determine the value of m*?
Response:
We have added the following explanation:
The value of is assumed to be the same for all temperatures. is calculated by the formula: which provides a reliable estimate for the effective mass obtained from Shubnikov-de Haas oscillations [Pudalov 2002]." where is the interaction parameter and the effective Bohr radius[32].
- The title should use "metal-insulator transition" instead of "MIT."
Response:
Done
- The addresses of the authors should be consistent throughout the manuscript.
Response:
Done
- In the abstract, what does "n2" refer to? This term should be defined or clarified.
Response:
According to your suggestion, we have clarified the term n2
- In the first paragraph of Introduction, "... study by Kravchenko et al. in 1994 [1-3] ... ": the study by Kravchenko et al. should be referenced from 1994 to 2001, not just 1994.
Response:
Done
- In the first paragraph of the introduction, it is mentioned that experimental studies were "carried out on Si-MOS samples [4]", but the term "MOS" is not defined.
Response:
According to your suggestion, Acronym MOS is defined as : Metal oxide semiconductors
- In the first paragraph of the introduction, the statement "Several experimental studies in the ..." lacks references.
Response:
According to your suggestion, we have added the ref [1-3]
- The manuscript lacks sufficient information about 2D Si-MOSFETs and Si-15 samples. It would be helpful to include basic information about these samples to aid reader understanding. Referring to another paper (ref 22) for this information is inconvenient.
Response:
Authors thank the reviewer for this recommendation. We have added the following paragraph:
The conductivity measurements were conducted using a 3He-cryostat, reaching temperatures as low as 0.3 K. The four-terminal AC technique was employed, wherein lock-in amplifiers detected signals within the frequency range of 13 to 17 Hz.
- The data in Figure 1 should be cited from Ref 22. Please include a reference to Ref 22 in the caption of Figure 1.
Response:
We have added the sentence: “Figure adapted from Ref [25]”
- In Figure 2, the vertical axis should be labeled as sigma0, not 00.
Response:
Done
- It is unclear how the authors fit the data to obtain conductivity. What method was used? Additionally, information about standard errors should be provided.
Response:
In two dimensional systems (2D), conductivity is the inverse of resistivity.
- In Figure 3, the symbol used for the slope on the vertical axis is incorrect.
Response:
(e2/h per K) is replaced by (e2/hK)
- The term "ns" is defined inconsistently, referring to it as surface density, carrier density, or electron density in different paragraphs.
Response:
ns is the electron density
- Several references (ref 13, 18, 20, 21, 22, 24, 33, 34) are not consistently formatted. Please ensure uniformity in referencing style throughout the manuscript
Response:
All Refs have been checked

Reviewer 2 Report
The authors investigated the mechanism of the metal-insulator transition (MIT) near the critical carrier density. Their results are interesting, but some points could be revised for the readers.
- In Figure 1, the authors conducted linear fitting of the conductivity data. The fit is good under the critical carrier density (ncs), but it is not as good at higher carrier densities, especially below 2.1 K. The conductivity appears to be independent of temperature below 2.1 K. The authors should provide more explanation of this behavior.
- The image in Figure 1 is cropped. The authors should check the image and make sure that it is complete.
- It could be helpful to describe the detailed method used to measure the conductivity at low temperatures.
- The authors should proofread Figure 2, Figure 3, and the manuscript for typos.
- The authors should emphasize their own model by comparing it to previously reported percolation models in more detail. This would help readers understand their model better.
Author Response
Dear Editor,
Thank you very much for handling our paper. I am pleased to send you our revised paper.
Please note that changes are written in blue.
Rviewer2: Dear Reviewer,
Thank you very much for your review. Your help improved the manuscript quality.
The authors investigated the mechanism of the metal-insulator transition (MIT) near the critical carrier density. Their results are interesting, but some points could be revised for the readers.
- In Figure 1, the authors conducted linear fitting of the conductivity data. The fit is good under the critical carrier density (ncs), but it is not as good at higher carrier densities, especially below 2.1 K. The conductivity appears to be independent of temperature below 2.1 K. The authors should provide more explanation of this behavior.
Response:
We have added the following explanation:
The conductivity appears to be independent of temperature below 2.1K. The percolation model adopted is only valid for densities very close to the critical density (outliers at very low temperatures are few in number compared with the total number of experimental points). This aberration is incomprehensible and unexpected. We suspect the appearance of quantum phenomena at very low temperatures. That's why we only consider linear points.
- The image in Figure 1 is cropped. The authors should check the image and make sure that it is complete.
Response:
We've checked the image and improved its visibility
- It could be helpful to describe the detailed method used to measure the conductivity at low temperatures.
Response:
Authors thank the reviewer for this recommendation. We have added the following paragraph:
The conductivity measurements were conducted using a 3He-cryostat, reaching temperatures as low as 0.3 K. The four-terminal AC technique was employed, wherein lock-in amplifiers detected signals within the frequency range of 13 to 17 Hz.
- The authors should proofread Figure 2, Figure 3, and the manuscript for typos.
Response:
Done
- The authors should emphasize their own model by comparing it to previously reported percolation models in more detail. This would help readers understand their model better.
Response:
Authors thank the reviewer for suggestion. This helped us to improve the analysis by adding the following paragraph:
Several authors have proposed phenomenological percolation models to describe the metal-insulator transition. Y. Meir's non-interacting model [37] describes transport in the 2D gas using a classical network of quantum resistors, representing tunneling events connecting populated carrier regions across depopulated regions. Another notable model is J. Shi et al.'s meta-percolation model [38], which is a quantum percolation model in a square lattice with finite phase coherence. Transport through the links occurs coherently, with a phase decoherence introduced in a certain fraction of these links. In real systems, this type of percolation may arise from a mixture of two phases with different densities, where the high-density phase represents an electron liquid and the low-density phase corresponds to a Wigner crystal or glass.
The percolation model proposed by He and Xie describes a system consisting of a conductive liquid phase separated by regions of an insulating vapor phase [39]. In the case of Si-MOSFETs, the negative charges of the electron gas are compensated by mobile positive charges from the gate. Thus, the system can be considered as two interacting carrier gases: the 2D electron gas and a hole gas. At high electron density, the system behaves like a metallic electron-hole liquid, while at low density, it behaves as an insulating exciton gas. At an intermediate density, there is a mixture of these two phases.
Shi et al. [39] proposed that an electron gas with rs > 2 becomes unstable, and it is favorable for it to reduce its surface area in order to increase rs. This leads to the formation of dense liquid droplets surrounded by empty regions (or regions of lower density, forming a disordered Wigner crystal when the temperature is non-zero). However, this reduction in surface area is accompanied by an increase in Fermi energy. They demonstrated that, in a system without disorder, the mechanism of droplet formation is unfavorable. On the other hand, in the presence of disorder, electrons tend to gather in potential minima, promoting droplet formation. They further showed that the Coulomb interaction needs to be screened for the formation of these droplets, which is the case in experimental systems, either due to a metallic gate in silicon MOSFETs or through the Thomas-Fermi mechanism in heterostructures.
It is worth noting that the model developed by Spivak [40] to interpret the behavior of resistivity in the metallic phase leads to a similar scenario of percolation, where a metallic Fermi liquid percolates within an insulating Wigner crystal. Spivak also considers the formation of liquid regions mixed with crystallites, which are stabilized by the screened Coulomb interaction between electrons. These various models are all based on the presence of density inhomogeneities in the metallic phase.

Round 2
Reviewer 1 Report
It is much better.
Author Response
Dear Reviewer,
Thank you very much for your review. Your help improved the manuscript quality.
